# The Fd4 transcription factor translates transient spatial cues in progenitors into long-term lineage identity

Sen-Lin Lai*, Chris Q Doe*

Institute of Neuroscience, Howard Hughes Medical Institute, University of Oregon, Eugene, United States

## eLife Assessment

This **important** study focuses on the molecular mechanisms underlying the generation of neuronal diversity. Taking advantage of a well-defined neuroblast lineage in Drosophila, the authors provide **convincing** evidence that two transcription factors of the conserved forkhead box (FOX) family offer a mechanistic link between transient spatial cues that specify neuroblast identity and terminal selector genes that define post-mitotic neuron identity. The findings will be of interest to developmental neurobiologists.

*For correspondence:
slai@uoregon.edu (S-LL);
cdoe@uoregon.edu (CQD)

Competing interest: The authors declare that no competing interests exist.

**Abstract** Neural diversity is required for the brain to generate complex behaviors. During development, neural progenitors are exposed to different combinations of transient spatial cues for their identity specification. This identity is then interpreted by their progeny to activate terminal selector genes to become lineage-specific neurons. After spatial cues fade, it remains unclear how progenitors maintain their unique identity so that their progeny express the accurate, lineage-specific terminal selector genes. Using single-cell RNA sequencing in *Drosophila*, we identified a Forkhead domain transcription factor, Fd4, that is exclusively expressed in a single neural progenitor (neuroblast) and its new-born progeny. This neuroblast (NB), named NB7-1, forms at the intersection of the transient spatial cues Vnd (columnar expression) and En (row expression). We show that Fd4 expression overlaps spatial factor expression and terminal selector gene expression, thereby making Fd4 an excellent candidate for bridging transient spatial factors to lineage-specific terminal selector genes. We show that Fd4 is required for expression of terminal selector genes that maintain neuronal identity. Conversely, Fd4 misexpression generates ectopic NB7-1 progeny at the expense of Fd4-negative progenitor lineages. We conclude that Fd4 is continuously expressed in the NB7-1 and its new-born neuronal progeny where it activates terminal selector genes to produce lineage-specific neurons. We propose that Fd4 is a pioneering member of a class of 'lineage identity genes' that translate transient spatial cues into a long-term lineage identity.

## Introduction

From flies to mammals, neurogenesis begins with the formation of a small population of neural stem cells that generate a large diversity of neurons necessary to generate complex behaviors. In the mammalian nervous system, Hox genes play a critical role in patterning and segmenting the rostro-caudal axis of the neural tube. Opposing gradients of morphogen Sonic hedgehog (Shh) and Wnt/BMP signaling then establish dorsoventral patterning. Within the domains shaped by the interaction of Hox genes and Shh/Wnt/BMP signals, homeobox genes including Pax, Nkx, and Dbx are expressed in spatially restricted patterns, subdividing the neural tube into distinct progenitor domains. Each of

**eLife digest** The brain contains a diverse range of neurons that form networks responsible for our thoughts, movements and behaviours. All neurons originate from neural stem cells, the earliest cells of the nervous system. However, neural stem cells are not identical; each is programmed to generate specific neuron types.

During early development, stem cells receive brief signals that provide temporary instructions about which neurons to produce. A key challenge is that these signals are short-lived, whereas brain development occurs over a much longer period. To overcome this, neural stem cells must "remember" their identity, converting transient signals into long-lasting cellular memory that ensures consistent neuron production.

Lai and Doe investigated how short-lived developmental signals are transformed into durable memory within neural stem cells. They asked whether a specific "memory molecule" preserves stem-cell identity and whether transferring this molecule to another stem cell could redirect it to produce a different neuron type. To find out how stem cells determine their fate and generate the correct neurons at the appropriate times, the researchers used genetically modified fruit flies.

The experiments identified the transcription factor Fd4 as a key molecule that maintains long-lasting cellular memory in neural stem cells. This protein is known to help control which genes are turned on and off inside a cell and helps determine what type of cell it becomes. In the flies, Fd4 was activated by developmental signals and remained expressed in specific stem cell lineages. When Fd4 (and its partner Fd5) was removed, the stem cells failed to produce the neurons they normally generate. Conversely, forcing other stem cells to express Fd4 redirected them to produce neurons typically associated with Fd4-expressing lineages. These results demonstrate that Fd4 preserves the identity of stem cells over time and guides the production of the correct neuron types throughout development.

These findings have important implications for research in brain development and neurological disorders, particularly for efforts to guide stem cells to generate specific neuron types. In the long term, this knowledge could contribute to regenerative therapies aimed at replacing damaged neurons. However, further research is needed to determine whether this mechanism exists in humans, including identifying equivalent genes and evaluating safe ways to manipulate them before clinical application.

these domains subsequently expresses distinct combinations of terminal selector genes that maintain neuronal identity (*Philippidou and Dasen, 2013*; *Sagner and Briscoe, 2019*).

In the *Drosophila* nervous system, Hox genes also pattern the anterior-posterior axis of the neuroepithelium (*Technau et al., 2014*). However, individual neuroblast identity within each segment is determined by the combinatorial action of 'spatial transcription factors' (STFs) expressed in rows and columns in every segment (*Skeath and Thor, 2003*). The columnar genes *ventral nervous system defective* (*vnd*), *intermediate neuroblast defective* (*ind*), and *muscle segment homeobox* (*msh*; FlyBase: *Dr*) subdivide the neuroepithelium into ventral, intermediate, and dorsal columns, respectively (*Isshiki et al., 1997*; *McDonald et al., 1998*; *Weiss et al., 1998*). Similarly, the row genes *mirror* (*mirr*), *hedgehog* (*hh*), *wingless* (*wg*), *gooseberry* (*gsb*), and *engrailed* (*en*) subdivide the neuroepithelium along the anterior-posterior axis. Together, the row and column genes subdivide neuroectoderm into a chessboard-like pattern (*Skeath and Thor, 2003*), with each 'square' of this pattern consisting of a unique STF combination. All three columnar genes function to specify neuroblast columnar identity (*Isshiki et al., 1997*; *McDonald et al., 1998*; *Weiss et al., 1998*); similarly, the row genes *en*, *hh*, *wg*, and *gsb* specify neuroblast row 1–7 identity (*Anderson et al., 2025*; *Bhat, 1996*; *Chu-LaGraff and Doe, 1993*; *Skeath et al., 1995*; *Zhang et al., 1994*). Subsequently, most neuroblasts sequentially express a cascade of 'temporal transcription factors' (TTFs) that diversify neurons in each neuroblast lineage (*Doe, 2017*; *Isshiki et al., 2001*; *Pollington et al., 2023*). The integration of STF and TTF expression activates downstream terminal selection genes, generating a diversity of neuronal types (*Gabilondo et al., 2016*; *Stratmann and Thor, 2017*; *Stratmann et al., 2016*).

After STFs establish neuroblast identity, their expression fades away prior to the expression of terminal selector genes, which are typically homeodomain (HD) TFs. HDTFs are widely conserved

proteins that are first expressed in new-born postmitotic neurons (*Doe and Thor, 2024*; *Hobert, 2016*; *Leung et al., 2022*). This raises the question: What factors act to bridge transient STF expression that specifies initial neuroblast identity to expression of lineage-specific HDTFs that consolidate and maintain neuronal identity? Here, we report on the expression and function of the transcription factor Fd4 (Flybase: fd96Ca). We show that it (i) is expressed continuously in NB7-1 during embryonic stages, (ii) is transiently expressed in new-born neurons in this lineage, (iii) is necessary to specify NB7-1 progeny identity, and (iv) is sufficient to induce ectopic NB7-1 identity. This places Fd4 as downstream of STFs but upstream of terminal selector genes. We propose that Fd4 acts as a 'neuroblast lineage identity gene' that acts as a bridge linking transient spatial cues that specify initial neuroblast identity to the terminal selector genes that maintain neuroblast progeny identity.

## Results

### Fd4 is transiently co-expressed with STFs and is maintained throughout the NB7-1 lineage

FD4 is specifically expressed in NB7-1 (*Anderson et al., 2025*) and first detectable in NB7-1 at stage 11, where it is co-expressed with En and Vnd STFs (*Figure 1A*, *Figure 1—figure supplement 1*). Expression of Fd4 in NB7-1 persists at least until the third instar larval stage (data not shown). In contrast, spatial factors En and Vnd are transiently expressed in the neuroectoderm prior to NB7-1 formation (*Chu et al., 1998*; *Jiménez et al., 1995*; *McDonald et al., 1998*; *Mellerick and Nirenberg, 1995*), in NB7-1 prior to Fd4 expression, and are downregulated by stage 13 (*Figure 1A and B*, *Figure 1—figure supplement 1*). Thus, En and Vnd expression precedes Fd4, followed by a window of co-expression, and then the STFs disappear and Fd4 maintains expression (*Figure 1C*). This leads to the following hypotheses: (i) STFs specify the neuroblast and its early-born progeny, (ii) STFs and Fd4 act redundantly to specify mid-born neurons in the lineage, and (iii) Fd4 alone maintains neuroblast identity and specifies late-born neurons in the lineage (*Figure 1D*).

### Fd4 is expressed in NBs, GMCs, and new-born neurons, but not in differentiated neurons

STFs are expressed in neuroblasts and GMCs, with little, if any, expression in postmitotic neurons (*Chen and Konstantinides, 2022*). In contrast, terminal selector genes (e.g. Even-skipped; Eve) are typically expressed in new-born postmitotic neurons (*Doe et al., 1988*). This raises the question: What factors act to bridge transient STF expression that specifies initial neuroblast identity to expression of lineage-specific HDTF terminal selectors that consolidate and maintain neuronal identity?

To determine the expression of Fd4 along the neuron differentiation axis (neuroblast>GMC>new-born neuron>mature neuron), we used an Fd4 antibody to examine its expression from neuroblast to neuron differentiation. We observe Fd4 protein in neuroblasts (Wor+), GMCs (Wor+Pros+), new-born neurons (Pros+ Elav+ adjacent to the neuroblast), but not in mature neurons (Pros+ Elav+ far from the neuroblast) (*Figure 2A*). We conclude that Fd4 expression initiates in NB7-1, is maintained in GMCs and new-born neurons, and is downregulated in mature, differentiated neurons. This expression pattern effectively exposes all new-born neurons in the lineage to transient Fd4 at a time when they have not yet consolidated their neuronal identity.

To further validate our observations, we examined Fd4 expression in identified neurons within the NB7-1 lineage. NB7-1 sequentially produces five Eve+ U motor neurons (UMNs) from first-born U1 motor neuron to fifth-born U5 motor neuron. Eve is a terminal selector gene, and its continuous expression is required to maintain neuronal activity and function (*Heckscher et al., 2015*). We find that Fd4 expression first becomes detectable at embryonic stage 10 (*Figure 2B*). Notably, Fd4 is strongly expressed in newly born Eve+ neurons and progressively declines as Eve+ neurons mature (*Figure 2B and C*). Thus, unlike Eve, Fd4 expression is transient in postmitotic neurons and restricted to the time immediately following their birth (summarized in *Figure 2D*). Taken together, we find that Fd4 is continuously expressed in NB7-1 but transiently expressed in new-born neurons. We propose that Fd4 is required to maintain NB7-1 lineage identity to prime new-born neurons to activate lineage-appropriate terminal selector genes (*Figure 2E*).

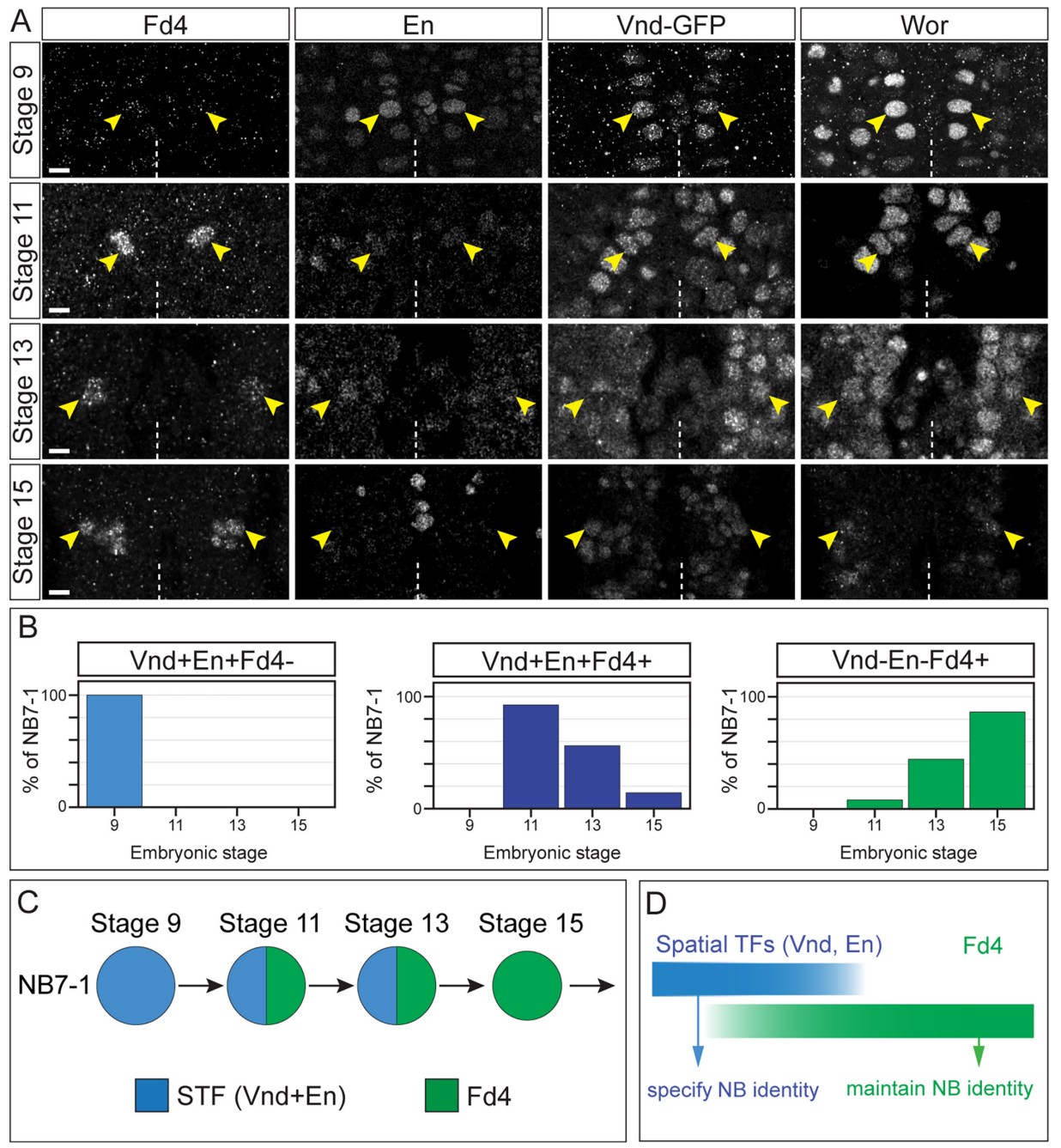

**Figure 1.** Fd4 is expressed in NB7-1 after the expression of spatial factors Vnd and En. (**A**) Expression of Fd4, En, Vnd, and Wor in a segment during embryonic stages 9, 11, 13, and 15. Anterior, up. Dashed lines, ventral midline. Scale bar: 10 µm. (**B**) Quantification of NB7-1 expressing various combinations of En, Vnd, and Fd4. (**C**) Summary of expression for Fd4, and En Vnd double-positive cells. STF: spatial transcription factors. (**D**) Proposed function of spatial transcription factors and Fd4 in neuroblast identity.

The online version of this article includes the following figure supplement(s) for figure 1:

**Figure supplement 1.** Fd4 expression follows the expression of spatial factors En and Vnd.

## Fd4 is required to specify NB7-1 progeny

Neuroblast identity is determined by its molecular profile; neuroblast *lineage* identity is determined by the neural progeny each neuroblast produces. How does Fd4 maintain NB7-1 lineage identity? To study this question, we assayed the production of NB7-1 progeny in *fd4* mutant and Fd4 misexpression paradigms. Within the NB7-1 lineage, early-born progeny express the terminal selector genes:

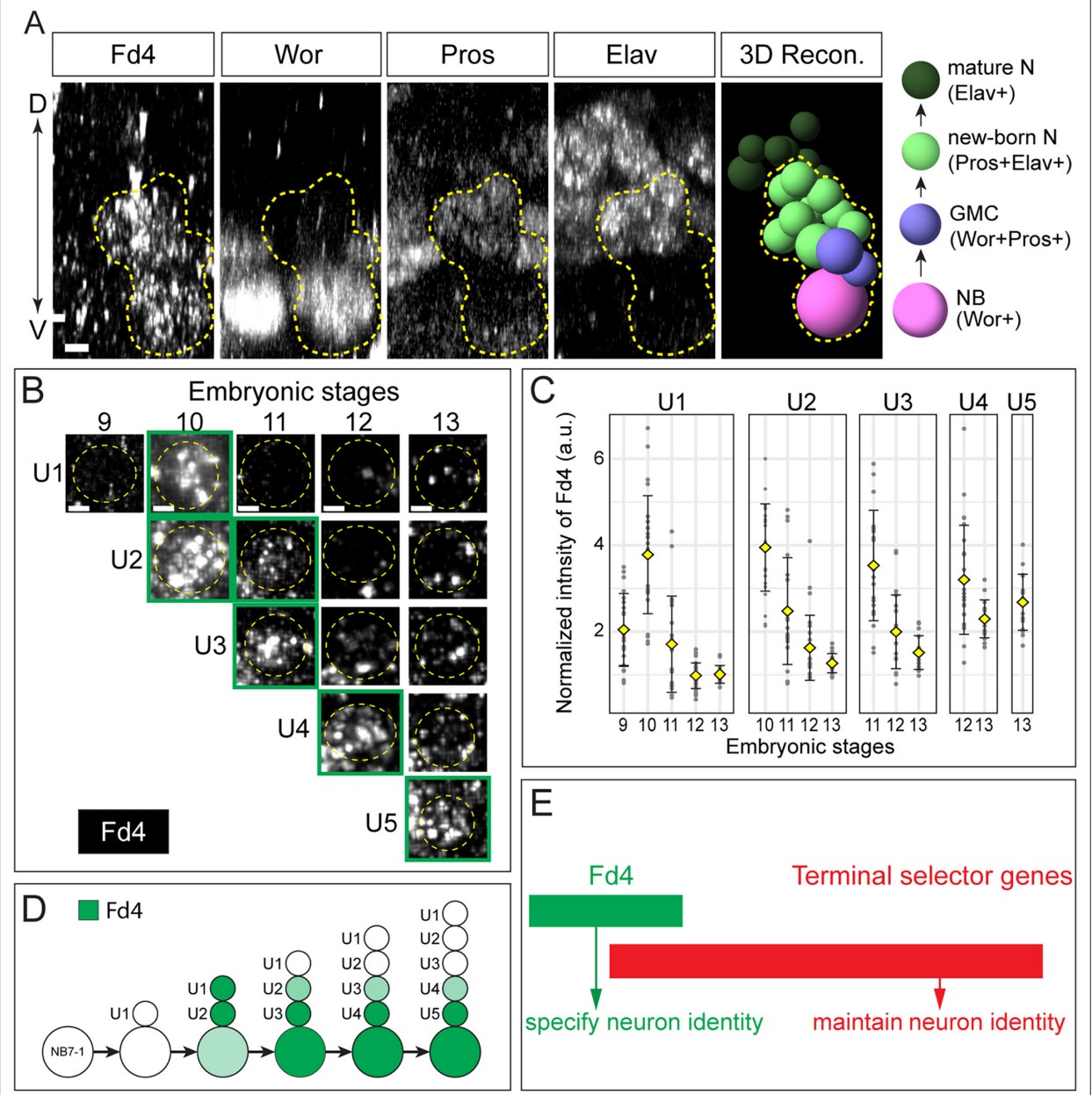

**Figure 2.** Fd4 is expressed in neuroblasts (NBs), GMCs, and new-born neurons, but not in differentiated neurons. (**A**) Expression of Fd4, Worniu (Wor), Prospero (Pros), and Elav in a hemisegment of a stage 12 embryo. Posterior view with dorsal oriented upward and ventral downward. The left four panels show marker expression, and the fifth panel (3D Recon.) shows a 3D reconstruction of Fd4+ cells and adjacent Fd4-Elav+ cells from the left four panels using Imaris Spots function. The rightmost image shows the expression profiles of different cell types. Yellow dashed lines outline the cells expressing Fd4. Scale bar: 2 µm. (**B**) Representative images of Fd4 expression in U motor neurons (UMNs) in the stage 9, 10, 11, 12, and 13 embryos. Yellow dashed lines outline the UMNs. Green squares box the UMNs that express FD4. Scale bar: 1 µm. (**C**) Quantification of Fd4 intensity in UMNs. Each dot is a measurement of Fd4 fluorescence intensity in a UMN normalized to the fluorescent intensity in an Fd4-negative cell. Yellow diamond represents the average intensity, and error bars are standard deviation. a.u., artificial unit. Number of hemisegments with UMNs measured: stage 9, 21; stage 10, 24; stage 11, 20; stage 12, 26; stage 13, 18. (**D**) Summary of Fd4 expression in Eve+ cells during early NB7-1 lineage. The intensity of green represents the expression levels of Fd4. (**E**) Proposed functions of Fd4 and terminal selector genes in neuron identity.

Eve (motor neurons) and Dbx (interneuron siblings) (*Lacin et al., 2009*). In total, eight Dbx+ inter-neurons are produced in this lineage: five are produced as siblings of Eve+ neurons, whereas the remaining three are born later, after Cas is expressed in NB7-1, and are Cas+ (see below).

In *fd4* mutants, we observe no change in the number of Eve+ neurons or Dbx+ neurons (n=40 hemisegments). However, the tandem gene *fd5* (Flybase: *fd96Cb*) is also enriched in NB7-1 lineage

(*Anderson et al., 2025*), acts redundantly with *fd4* during leg development (*Ruiz-Losada et al., 2021*), and is co-expressed with *fd4* in NB7-1 and its progeny (*Figure 3—figure supplement 1*), suggesting they act redundantly in the NB7-1 lineage. Not surprisingly, we found that *fd4* single mutants or *fd5* single mutants had no phenotype (Eve+ neurons were all normal). Thus, to assess their roles, we generated an *fd4* and *fd5* double mutant. Because many Eve+ and Dbx+ cells are generated outside of NB7-1 lineage, it was also essential to identify the Eve+ or Dbx+ cells within NB7-1 lineage in wild-type and *fd4/fd5* mutant embryos. To achieve this, we replaced the open reading frame (ORF) of *fd4* with *gal4* (called *fd4-gal4*) (see Methods); this stock simultaneously knocks out both *fd4* and *fd5* (called *fd4/fd5* mutant hereafter) while specifically labeling the NB7-1 lineage. For the remainder of this paper, we use the *fd4/fd5* double mutant to assay for loss-of-function phenotypes. We find that in wild type, NB7-1 generates 28.6±2.6 cells, while *fd4/fd5* mutant produces 27.0±5.5 cells (*Figure 3A*; quantified in *Figure 3B*). Thus, Fd4 is not required for NB7-1 proliferation.

In contrast, we observe a significant loss of Eve+ and Dbx+ cells in *fd4/fd5* mutant embryos (*Figure 3A*; quantified in *Figure 3C and D*). Further analysis shows that the missing Eve+ cells are later-born Runt+ U4-U5 neurons (*Figure 3E*; quantified in *Figure 3F*) and their corresponding body wall muscle targets are also missing (*Figure 3—figure supplement 2*). The missing Dbx+ cells are later-born Cas+ interneurons, even though the overall number of Cas+ cells remains unchanged (*Figure 3G*; quantified in *Figure 3H and I*). Taken together, we find that early U1-U3 neurons are generated independently of Fd4, whereas later-born Runt+ U4-U5 and Cas+Dbx+ interneurons require Fd4 for their proper specification.

The early-born cells were unaffected in the *fd4/fd5* mutant, raising the possibility that these neurons could be directly specified by the integration of spatial factors En and Vnd (*Figure 1*), independent of Fd4 and Fd5. To test this hypothesis, we used the *en-gal4* driver to express *UAS-vnd* in the *fd4/fd5* mutant background. We found more Eve+ cells per hemisegment than *fd4/fd5* mutant alone (*Figure 3J*). In addition, 0.2±0.5 Eve+ cells were ectopic Hb+ (excluding U1/U2), indicating that Vnd-En integration is sufficient to generate both early-born and late-born Eve+ cells in the *fd4/fd5* mutants. We conclude that the integration of Vnd-En specifies NB7-1 identity, and Fd4 acts to maintain NB7-1 identity and specify late-born neurons (*Figure 3K*).

## Fd4 misexpression induces ectopic NB7-1-specific progeny

We have shown that Fd4 is required for proper generation of NB7-1 progeny, raising the question of whether Fd4 is sufficient to induce ectopic NB7-1 progeny in other neuroblast lineages. To determine if other lineages could be transformed into the NB7-1 lineage, we misexpressed Fd4 using the *sca-gal4* driver, which is first expressed in all neuroectoderm and persists into all newly formed neuroblasts, and assayed for Eve+ Dbx+ neurons (see above). In wild type, each abdominal hemisegment produced 18.1±1.2 Eve+ cells and 17.2±1.8 Dbx+ cells, including those in the NB7-1 lineage (*Figure 4A and B*). In contrast, pan-neuroblast expression of Fd4 resulted in widespread expression of the NB7-1 lineage markers Eve and Dbx in all regions of the hemisegment (*Figure 4C and D*; quantified in *Figure 4E and F*). In addition, we found the proportion of early-born (Eve+ Hb+) cells is slightly reduced, but the proportion of late-born cells (Eve+ Runt+) remains similar (*Figure 4—figure supplement 1*). Notably, misexpression of Fd5 didn't induce any NB7-1 lineage markers (data not shown). Our results support a model in which Fd4 is sufficient to induce NB7-1 lineage identity in most, if not all, lineages.

We next asked whether widespread expansion of Eve+ and Dbx+ cells following Fd4 misexpression was due to altered spatial patterning (*Figure 3A*) or altered temporal patterning (*Figure 4—figure supplement 2E*). We misexpressed Fd4 in the neuroectoderm and found no change in the expression of Vnd or En (*Figure 4—figure supplement 2B and C*), of Ind (*Figure 4—figure supplement 2A*) or of Wg (*Figure 4—figure supplement 2D*). Furthermore, continuous misexpression of Fd4 in neuroblasts from the Hb to Cas temporal window did not affect the timing of early (Hb) or late (Cas) TTF expression (*Figure 4—figure supplement 2F*). We conclude that Fd4 does not regulate STF or TTF patterning, and importantly, that Fd4 activates terminal selector genes Eve and Dbx in the NB7-1 lineage.

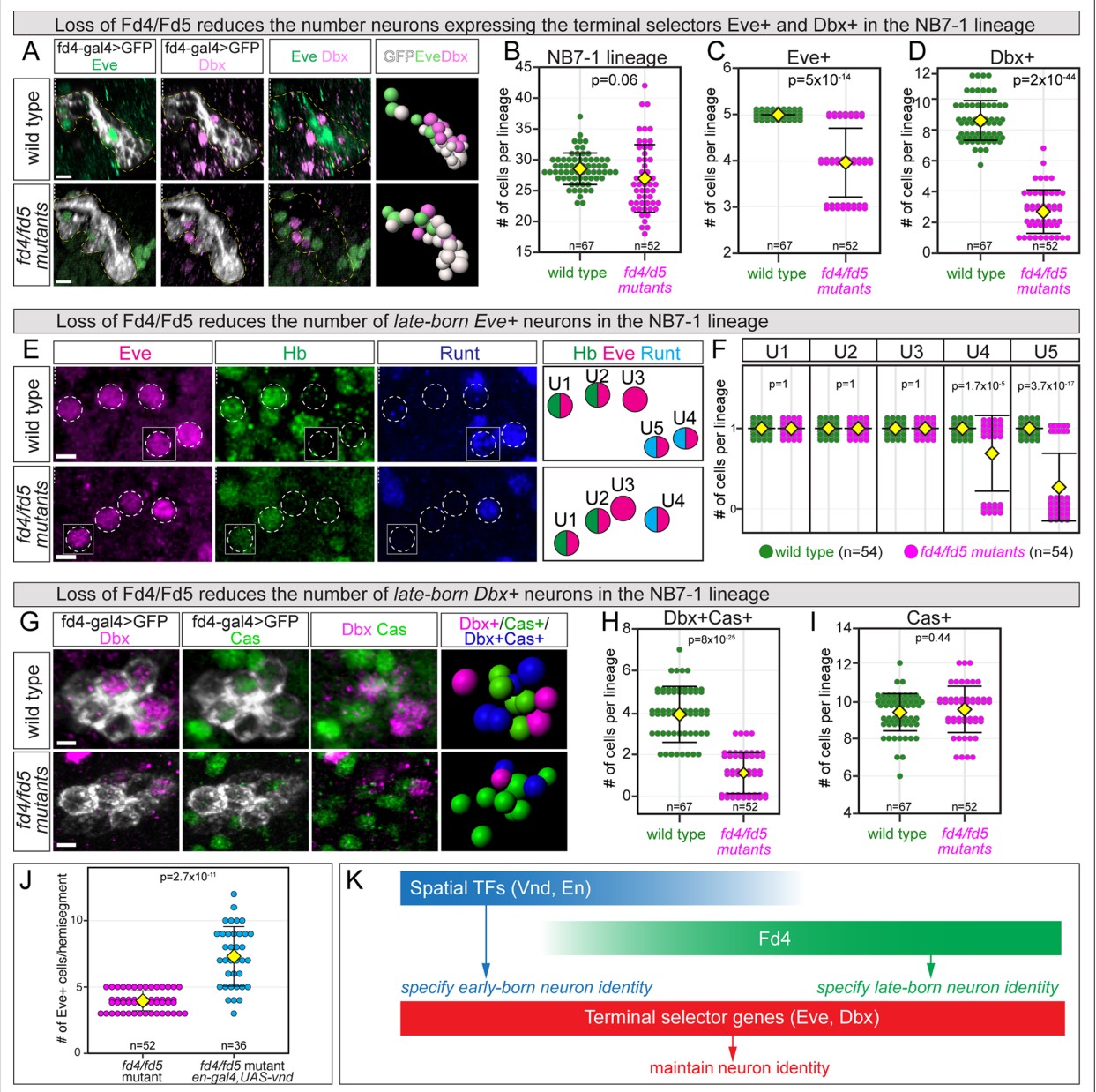

**Figure 3.** Fd4 is required for late-born neuron specification. (**A**) Eve and Dbx expression in the NB7-1 lineage of wild-type and *fd4/fd5* mutant stage 17 embryos. Posterior view with dorsal oriented upward and ventral downward. The NB7-1 lineage is outlined with a yellow dashed line. Rightmost panels are 3D reconstruction of the NB7-1 lineage from the left three panels with Imaris Spots function. Scale bar: 2 μm. (**B–D**) Quantification of total (**B**), Eve+ (**C**), and Dbx+ (**D**) cells in each lineage. Each dot represents an individual lineage. Yellow diamond, mean; error bars, standard deviation; n, number of lineages analyzed; p, the p-value of Student's t-test. (**E**) UMNs in wild type and *fd4/fd5* mutant stage 17 embryos. The identity of UMNs is determined by the expression of marker Hb or Runt and the relative position of the cell within the hemisegment. The rightmost panels are summary cartoons from the left panels. Scale bar: 2 μm. (**F**) Quantification of each individual UMNs in each lineage. Each dot represents an individual lineage. Yellow diamond, mean; error bars, standard deviation; n, number of lineages analyzed; p, the p-value of Student's t-test. (**G**) Dbx and Cas expression in the NB7-1 lineage of wild-type and *fd4/fd5* mutant stage 17 embryos. Dorsal view with anterior oriented upward and posterior downward. Rightmost panels are 3D reconstruction of the NB7-1 lineage from the left three panels with Imaris Spots function. Scale bar: 2 μm. (**H–I**) Quantification. Each dot represents an individual lineage. Yellow diamond, mean; error bars, standard deviation; n, number of lineages analyzed; p, the p-value of Student's t-test. Genotypes (**A–I**): wild type: *fd4-gal4,UAS-myr-sfGFP; fd4/fd5* mutant: *fd4-gal4,fd5¹ⁿᵗ/Df(3R)BSC493, UAS-myr-sfGFP*. (**J**) Quantification of Eve+ cells in *fd4/fd5* mutant (the same data as **C**) and vnd misexpression with en-gal4 in *fd4/fd5* mutant background. Each dot represents an individual lineage. Yellow diamond, mean; error bars, standard deviation; n, number of lineages analyzed; p, the p-value of Student's t-test. Genotype: *fd4/fd5* mutant, *fd4-gal4,fd5¹ⁿᵗ/Df(3R)BSC493*. (**K**) Summary of functions of spatial TFs, Fd4, and terminal selector genes in neuron identity.

The online version of this article includes the following figure supplement(s) for figure 3:

*Figure 3 continued on next page*

*Figure 3 continued*

**Figure supplement 1.** *fd4* and *fd5* co-express in the ventral nerve cord during embryogenesis.

**Figure supplement 2.** Loss of fd4/fd5 leads to the loss of corresponding muscle targeting.

## Fd4 misexpression induces NB7-1 lineage markers and represses NB5-6 lineage markers

To more precisely explore the effects of Fd4 in individual neuroblast lineages, we selectively misexpressed Fd4 in NB5-6 (this section) and NB7-3 (next section), as we have excellent lineage markers for both neuroblasts. In wild type, NB5-6 delaminates simultaneously with NB7-1, but is located at the lateral side of the neuroblast array and is specified by spatial factors Wg (row anterior to En) and Msh (lateral column) (*Chu-LaGraff and Doe, 1993*; *Isshiki et al., 1997*). The NB5-6 lineage and its progeny can be specifically labeled with *lbe-Gal4* (*Figure 5A*; *Baumgardt et al., 2009*). The thoracic NB5-6 (NB5-6T) generates 20.2±2.9 cells (*Figure 5B*) before undergoing apoptosis and does not produce Eve+ U1-U5 MNs (*Figure 5C*). The last four cells produced by NB5-6T are specified by the LIM-HDTF Apterous (Ap) (*Baumgardt et al., 2009*; *Figure 5C*), which is never observed in the NB7-1 lineage. When Fd4 is misexpressed in the NB5-6 lineage using *lbe-Gal4*, the number of progeny remains the same (21.1±3.1) (*Figure 5A and B*), showing that Fd4 does not alter lineage length.

Although misexpression of Fd4 in NB5-6 does not alter lineage length, it results in a significant reduction in the number of the NB5-6 lineage marker Ap+ cells and a concomitant increase in the NB7-1 lineage marker Eve+ cells (*Figure 5C*). Moreover, the ectopic Eve+ cells express the early-born (Hb) and late-born (Runt) U1-U5 markers (*Figure 5D and E*). These ectopic Eve+ cells, like the wild-type U1-U5 MNs, project axons out of CNS and target the dorsal muscles (*Figure 5F and G*). When we dissected newly hatched larvae and stained for the motor neuron marker phosphorylated-Mad (pMad), we found that the ectopic Eve+ cells are pMad+, consistent with an induction of the Eve+ U1-U5 motor neurons (*Figure 5H*). Interestingly, following Fd4 misexpression, only 7 of 18 examined NB5-6 lineages produced Dbx+ cells (0.4±0.8), and just one of these was Cas+ (0.06±0.2); the reason for the difference in Eve and Dbx in response to Fd4 misexpression remains unknown. Taken together, we find that expression of Fd4 in NB5-6 is sufficient to transform NB5-6 lineage into NB7-1 lineage. We conclude that Fd4 is necessary and sufficient to induce NB7-1 lineage identity within the neuroblast population as a whole and specifically in NB5-6.

## Fd4 misexpression induces NB7-1 lineage markers and represses NB7-3 lineage markers

To determine if Fd4 could reprogram another neuroblast into NB7-1-like lineage, we misexpressed Fd4 in NB7-3, a neuroblast that is different from NB5-6 in many aspects. NB7-3 is one of the last neuroblasts to form (*Broadus et al., 1995*), has a relatively small size, generates a short three-division lineage, generates a ventral-muscle targeting motor neuron (GW), two serotonergic neurons (EW1, EW2), and one Corazonin+ cell (EW3) (*Karcavich and Doe, 2005*; *Novotny et al., 2002*), which are never observed in the NB7-1 lineage. The NB7-3 lineage can be labeled with *eagle-Gal4,* and its progeny can be identified with antibody staining against Serotonin (EW1, EW2) or Corazonin (EW3) (*Figure 6A*). When Fd4 was misexpressed in NB7-3 with *eagle-Gal4*, there was no change in NB7-3 lineage length producing an average of four neurons in control and misexpression experiments (*Figure 6B*, quantified in *Figure 6C*). Despite the unchanged neuron numbers, we observed a significant reduction of Serotonin+ and Corazonin+ neurons and a corresponding increase in Eve+ neurons (*Figure 6C*) plus a slight increase in Dbx+ neurons (wild type: 0 Dbx+ neurons; misexpression 0.1±0.4 [n=64 hemisegments] Dbx+ neurons). Serotonin/Corazonin and Eve+ neurons are mutually exclusive; we never observe cells co-expressing both markers, ruling out a mixed lineage identity. We conclude that Fd4 is sufficient to repress NB7-3-specific markers and activate NB7-1-specific markers.

To gain a deeper understanding of the role of Fd4 in specifying lineage identity, we assayed for changes in motor neuron projections in the NB7-1 and NB7-3 lineages. In wild type, the NB7-3 generates a GW motor neuron that co-expresses the ventral muscle motor neuron transcription factor Nkx6 (Flybase: HGTX) and the pan-motor neuron marker pMad (*Figure 7A*; quantified in *Figure 7B*). In contrast, NB7-1 generates the Eve+ pMad+ U1-U5 motor neurons. All five UMNs project to the dorsal body wall muscles (*Landgraf et al., 1997*; *Schmid et al., 1999*). When Fd4 is misexpressed in the

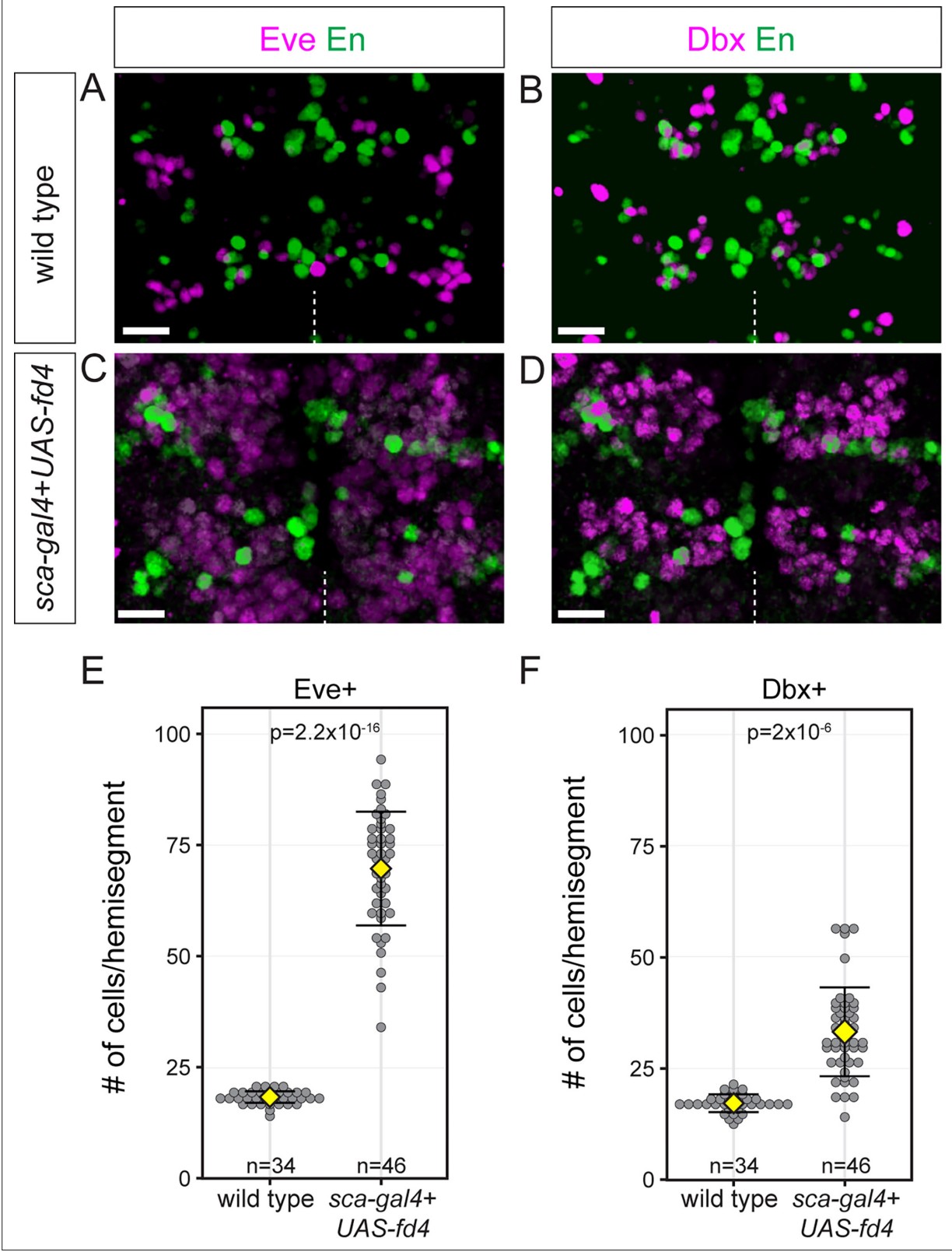

**Figure 4.** Widespread expression of NB7-1 lineage markers Eve and Dbx following ubiquitous Fd4 misexpression. (**A–B**) Eve and Dbx expression in wild type. En as a marker for segment boundary. Scale bars: 10 μm. (**C–D**) Eve and Dbx expression following Fd4 misexpression. En as a marker for segment boundary. Scale bars: 10 μm. (**E–F**) Quantification of Eve+ and Dbx+ cells in each hemisegment. Each dot represents an individual hemisegment. Yellow

*Figure 4 continued on next page*

*Figure 4 continued*

diamond, mean; error bars, standard deviation; n, number of lineages analyzed; p, the p-value of Student's t-test. Genotypes: wild type: *y-w-*; Fd4 misexpression: *sca-gal4,UAS-fd4*.

The online version of this article includes the following figure supplement(s) for figure 4:

**Figure supplement 1.** Ubiquitous Fd4 misexpression expands Eve+ neurons across both early- and late-born neurons.

**Figure supplement 2.** Fd4 does not regulate spatial patterning genes or temporal transcription factors.

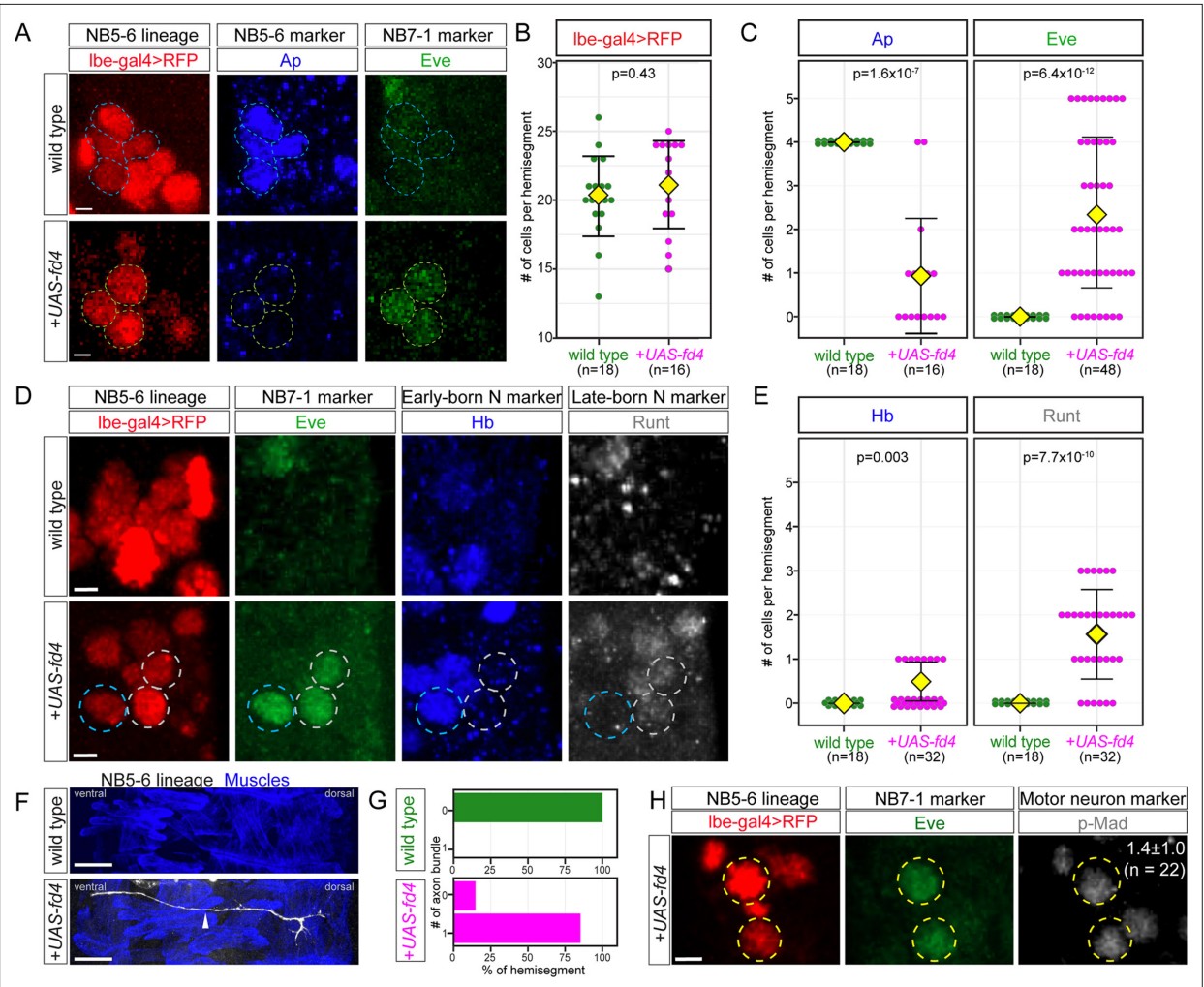

**Figure 5.** Fd4 misexpression in NB5-6 induces NB7-1 lineage markers and represses NB5-6 lineage markers. (**A**) Expression of Ap and Eve in wild type (top row) or following Fd4 misexpression (bottom row) in NB5-6 using the NB5-6-specific *lbe-gal4* driver. Scale bar: 2 µm. (**B–C**) Quantification of the number of RedStinger+ (**B**), Ap+ (**C**), and Eve+ (**C**) cells. Each dot represents an individual lineage. Yellow diamond, mean; error bars, standard deviation; n, number of lineages analyzed; p, the p-value of Student's t-test. (**D**) Expression of NB7-1 markers (Hb, Runt, Eve) in wild-type NB5-6 lineage and following Fd4 misexpression in the NB5-6 lineage. Scale bar: 2 µm. (**E**) Quantification of Hb+ Eve+ and Runt+ Eve+ cells. Each dot represents an individual lineage. Yellow diamond, mean; error bars, standard deviation; n, number of lineages analyzed; p, the p-value of Student's t-test. (**F**) Lateral view of lbe-gal4+ axon projection in wild type (top panel) and Fd4 misexpressed (bottom panel) embryos. The axons (white, arrowhead) were overlaying the body wall muscles (blue), and the muscles were labeled with antibody against Tropomyosin 1 (Tm1). Scale bars: 20 µm. (**G**) Quantification of percent of axons projected out of CNS to the body wall muscles in wild-type and Fd4 misexpressed embryos. (**H**) Expression of motor neuron marker pMad in ectopic Eve+ cells in Fd4 misexpressed NB5-6 lineage in newly hatched larvae. The number in the pMad panel shows the average number of pMad+ Eve+ cells per hemisegment. Scale bar: 2 µm. Genotypes: (**A–E, H**) wild type: *lbe-gal4,UAS-RedStinger*; Fd4 misexpression (*+UAS-fd4*): *lbe-gal4,UAS-RedStinger,UAS-fd4*; (**F–G**) wild type: *10xUAS-myr-smGdP.HA,lbe-gal4*. Fd4 misexpression (*+UAS-fd4*): *10xUAS-myr-smGdP.HA, lbe-gal4, UAS-fd4.*

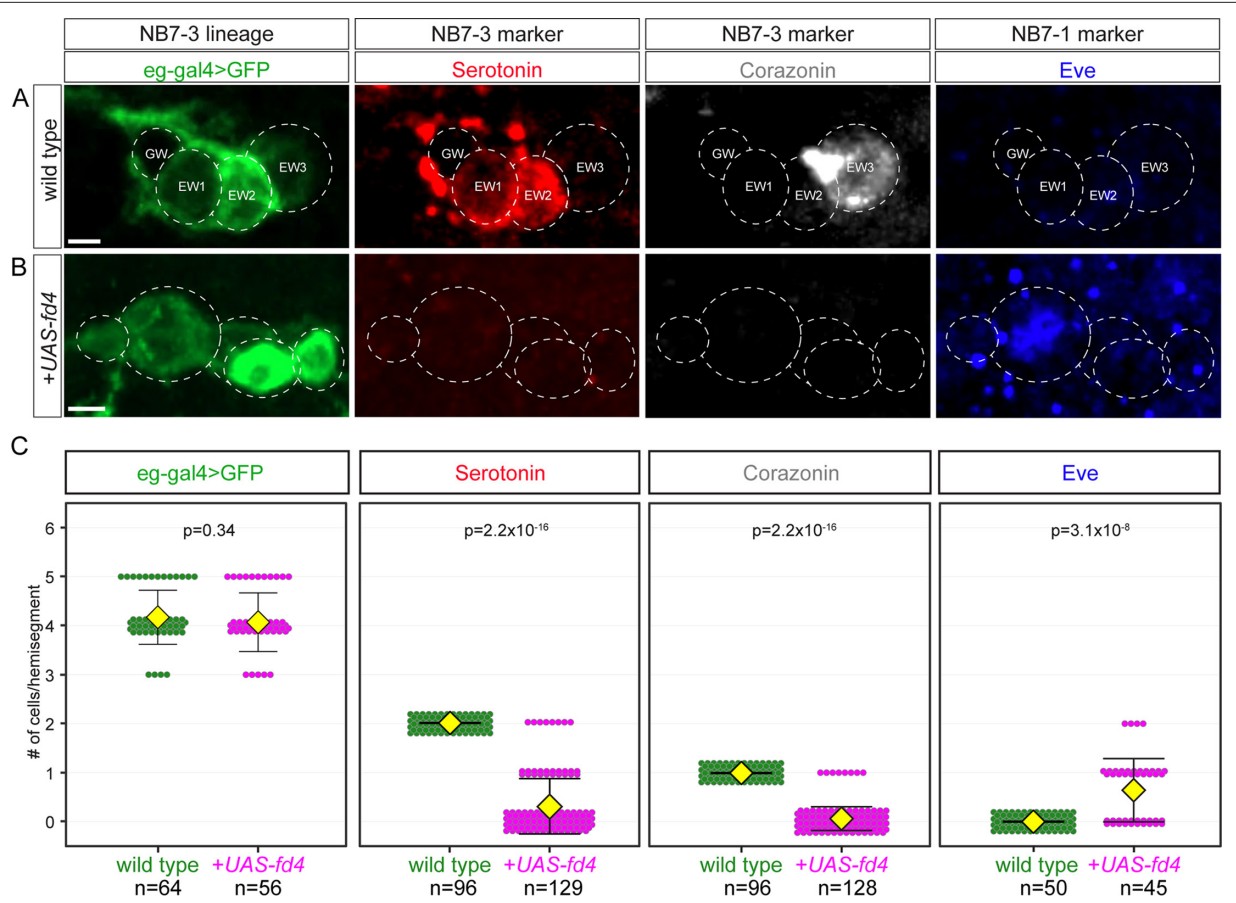

**Figure 6.** Fd4 misexpression in NB7-3 induces NB7-1 lineage markers and represses NB7-3 lineage markers. (**A–B**) Expression of serotonin (5-HT), Corazonin (Crz), and Eve in wild type (**A**) and following Fd4 misexpression in the NB7-3 lineage (**B**). Scale bar: 2 μm. (**C**) Quantification. Each dot represents an individual lineage. Yellow diamond, mean; error bars, standard deviation; n, number of lineages analyzed; p, the p-value of Student's t-test. Genotypes: wild type: *eg-gal4,UAS-myr-sfGFP*; Fd4 misexpression (*+UAS-fd4*): *eg-gal4,UAS-myr-sfGFP,UAS-fd4*.

NB7-3 lineage, we observed an increase in ectopic Eve+ motor neurons and a reduction in Nkx6+ motor neurons (*Figure 7A*; quantified in *Figure 7B*), indicating a transformation from NB7-3 to NB7-1 lineage identity. Interestingly, Fd4 misexpression in NB7-3 generates ectopic U1-U5 motor neurons that did not always fasciculate together when exiting the CNS (*Figure 7C*; quantified in *Figure 7D*); this indicates either incomplete U neuron specification or a difference in the timing of ectopic U neuron outgrowth. Furthermore, in wild type, the NB7-3-derived Nkx6+ motor neuron innervates a ventral body wall muscle, whereas NB7-1-derived Eve+ neurons innervate more dorsal body wall muscles (*Figure 7E*; quantified in *Figure 7G*; *Schmid et al., 1999*). In contrast, Fd4 misexpression in the NB7-3 lineage generated motor neurons that projected dorsally beyond their normal ventral muscle target (*Figure 7F*; quantified in *Figure 7G*; summarized in *Figure 7H*). We observed that these transformed neurons did not innervate the dorsal muscles. Perhaps their late birth did not give them time to extend to the most distant dorsal muscles, or they were incompletely specified. We conclude that Fd4 is sufficient to induce NB7-1 lineage identity at the expense of NB7-3 identity.

## Discussion

### Fd4 maintains neuroblast identity established by transient spatial factors En and Vnd

We identified Fd4 as an NB7-1-specific transcription factor which is continuously expressed in NB7-1 and its new-born neurons into larval stages. NB7-1 expresses En and Vnd, and loss of these spatial factors leads to the loss of NB7-1 (*McDonald et al., 1998*), loss of Fd4 (*Anderson et al., 2025*), and

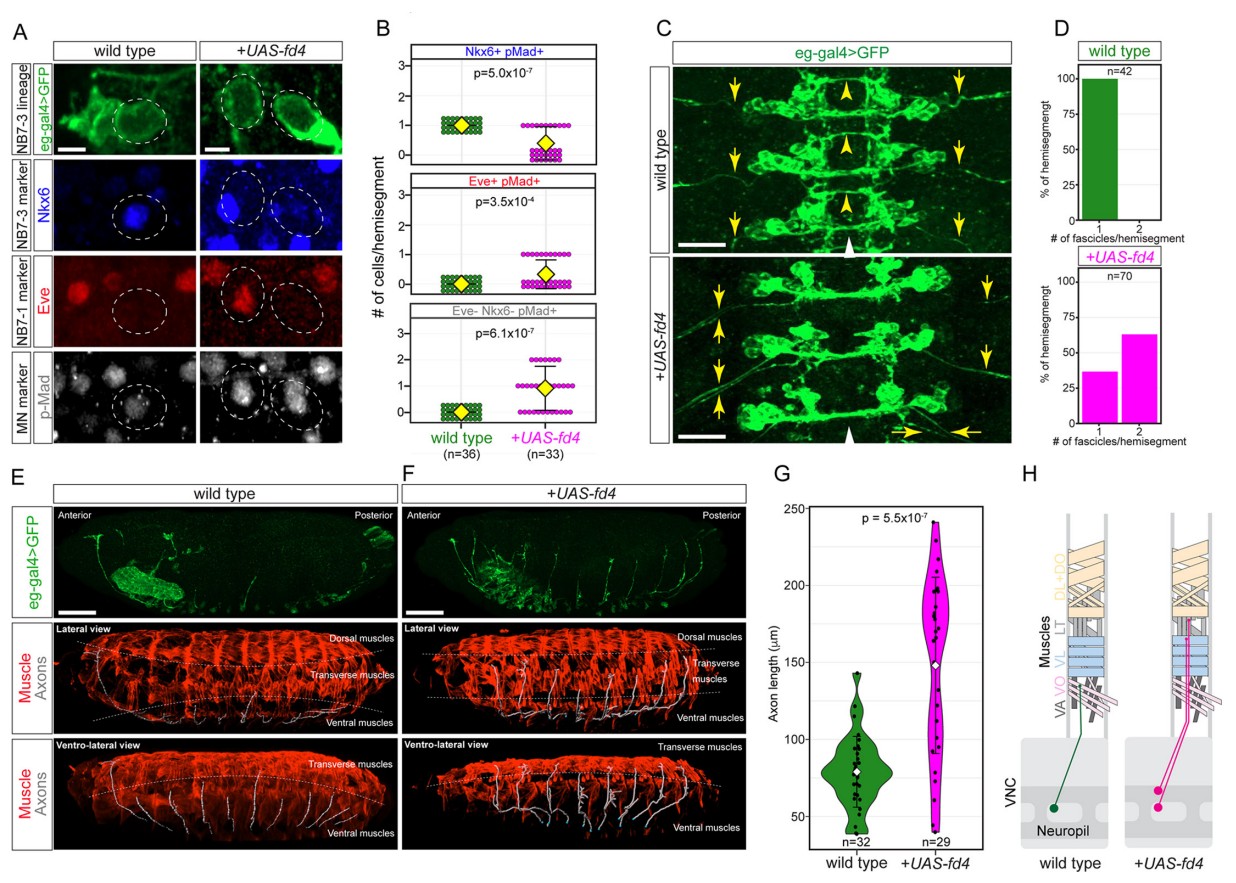

**Figure 7.** Fd4 misexpression in NB7-3 lineage dorsalizes motor neuron projections. (**A**) Expression of motor neuron markers Nkx6, Eve, and pMad in wild type (left column) and Fd4 misexpressed (right column) motor neuron in newly hatched larvae. Scale bars: 2 µm. (**B**) Quantification. Each dot represents an individual lineage. Yellow diamond, mean; error bars, standard deviation; n, number of lineages analyzed; p, the p-value of Student's t-test. (**C**) Dorsal view of three segments of wild type (top panel) and Fd4 misexpressed (bottom panel) neuronal projections. Yellow arrows indicate the fascicles projecting out from the neuropil. White arrowheads, ventral midline. Scale bars: 20 µm. (**D**) Quantification of fascicles exiting nervous system. Wild type (1±0; top panel). Fd4 misexpression (1.6±0.5; bottom panel). (**E, F**) Lateral view of *eg-gal4+* motor neuron axon projection in wild type (**E**; left panels) and Fd4 misexpressed (**F**; right panels) embryos. Top two panels are the maximum projections of confocal image stacks of *eg-gal4+* neurons. Middle and bottom panels are *eg-gal4+* neuron axons reconstructed with Imaris (white), overlaying the body wall muscles (red). The muscles are labeled with antibody against Tropomyosin 1 (Tm1). The dashed lines indicate the boundary between dorsal and longitudinal (middle panels), and longitudinal and ventral muscles (middle and bottom panels). Scale bars: 50 µm. (**G**) Quantification of motor neuron axon lengths. Each black dot represents the length of an axon measured from the VNC. White diamonds indicate the average length. The error bars are standard deviation. (**H**) Summary. Genotypes: wild type: *eg-gal4,UAS-myr-sfGFP*; Fd4 misexpression (*+UAS-fd4*): *eg-gal4,UAS-myr-sfGFP,UAS-fd4*.

loss of NB7-1 lineage markers (**McDonald et al., 1998**). Importantly, loss of *fd4/fd5* results in the loss of NB7-1 identity based on failure to generate the NB7-1-specific UMNs, whereas misexpression of Fd4 transforms most or all lineages toward an NB7-1 lineage. Thus, Fd4 is necessary and sufficient to specify lineage identity. We propose a model where transient expression of spatial factors En and Vnd activates Fd4 and establishes NB7-1 identity, with Fd4 translating transient spatial cues into a long-term lineage identity (**Figure 8**).

How does Fd4 sustain the positional identity established by spatial factors? In *Drosophila*, spatial factors regulate chromatin status, allowing temporal factors to bind to lineage-specific open chromatin and produce lineage-specific progeny (**Sen et al., 2019**). Thus, Fd4 may act downstream of En/Vnd to maintain chromatin status necessary for NB7-1-specific neuronal identity (e.g. Eve expression). Interestingly, Fd4 is also sufficient to activate NB7-1 lineage markers in other neuroblast lineages, suggesting Fd4 alone may be sufficient to change chromatin status, similar to other Forkhead domain proteins (**Guo et al., 2024**; **Jin et al., 2020**; **Sanese et al., 2019**). In mammals, other mechanisms for stabilizing lineage identity have been reported, including morphogens, transcriptional feedback

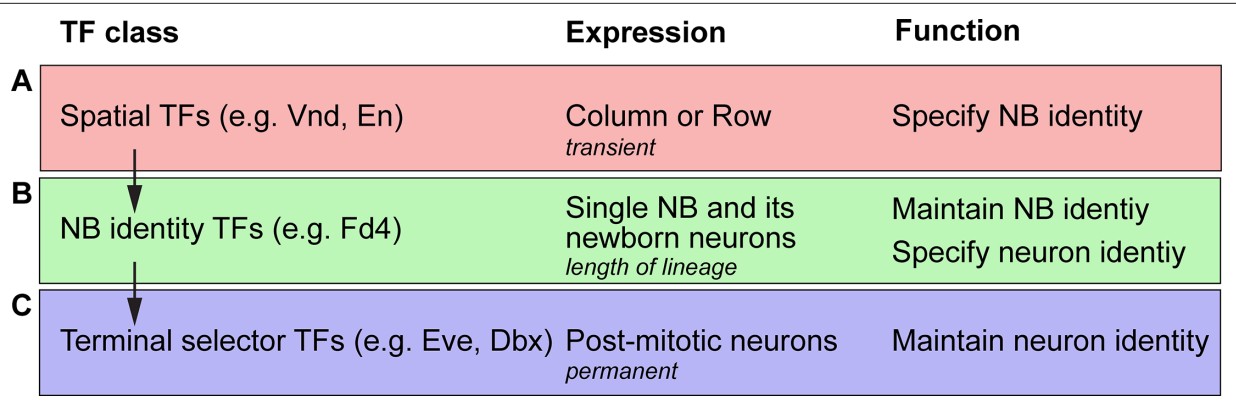

**Figure 8.** Model. We propose a three-step model for the specification and maintenance of neuroblast and neuron identity. (**A**) Spatial transcription factors (e.g. Vnd, En) are expressed transiently in rows and columns of neuroectoderm where they act combinatorially to specify neuroblast identity. (**B**) Neuroblast identity transcription factors (e.g. Fd4 or Lbe) are expressed in single neuroblasts and their new-born progeny throughout their lineage and act downstream of spatial factors to maintain neuroblast identity and upstream of terminal selector genes to specify late-born progeny. (**C**) Terminal selector genes (e.g. Eve, Dbx) are expressed in neurons where they permanently maintain neuron identity and functions.

mechanisms, noncoding RNAs, and chromatin regulators (reviewed in *Delgado and Lim, 2017*), and it remains an open question whether Fd4 uses any of these mechanisms in sustaining neuroblast identity. Interestingly, the *fd4/fd5* mutant maintains expression of *fd4:gal4*, suggesting that the *fd4/fd5* locus may have established a chromatin state that allows 'permanent' expression in the absence of Vnd, En, and Fd4/Fd5 proteins.

We found that the *fd4/fd5* mutant resulted in the loss of the NB7-1-specific U4 and U5 neurons (born after U1-U3) and late-born Dbx+ cells. Why does the loss of Fd4/Fd5 cause only a loss of late-born neurons? We suggest that the U1-U3 identities are specified by the STFs Vnd and En, which are expressed in the NB7-1 lineage during the time U1-U3 are produced (*Figure 1*, *Figure 1—figure supplement 1*). We propose that the STFs specify the early-born U1-U3 neurons, followed by Fd4/Fd5 taking a 'bridging' role of specifying the later-born U4/U5 neurons and maintaining all aspects of the lineage. This model would also explain the evolutionary advantage of genes like Fd4/Fd5. Specifically, if the overlapping expression of En and Vnd specifies NB7-1 identity, why the need for Fd4/Fd5? Given the transient nature of spatial cues, Fd4/Fd5 likely serve as a molecular 'memory' that preserves neuroblast identity, allowing late-born progeny to inherit lineage-specific transcriptional programs even after the original spatial cues have faded.

## Fd4 transforms lineage identity, but not lineage length

We found that Fd4 is sufficient to induce NB7-1 identity in NB7-3, but not sufficient to change the length of its lineage. NB7-3 is a late-forming neuroblast that has a small size and makes a short three-division lineage. Misexpression of Fd4 in NB7-3 was sufficient to activate NB7-1-specific lineage markers and repress NB7-3-specific lineage markers, but the NB7-3 lineage remained short. Thus, Fd4 can transform some aspects of neuroblast identity (molecular markers) but not all aspects (lineage length). It is likely that the smaller size of NB7-3 limits its number of divisions, or alternatively, unknown spatial factors may determine neuroblast lineage length.

## Fd4, Fd5 redundancy

Redundancy of closely related genes is fairly common in *Drosophila* (*Bhat and Schedl, 1997*; *Grosskortenhaus et al., 2006*; *Kohwi et al., 2011*; *Yeo et al., 1995*). In our studies, we found that the *fd4/fd5* double mutant lacks the late-born U4-U5 MNs (*Figure 4*); single mutants have no phenotype. Our misexpression experiments show that Fd4 alone is sufficient to promote NB7-1 identity (*Figures 5–8*). Fd5 alone has no ability to activate Fd4 or generate ectopic NB7-1-derived neurons (data not shown), indicating that the two genes are not fully redundant; it remains unclear why Fd4 plays the major role in transforming neuroblast identity. Based on the partial co-expression of *fd4* and *fd5* during embryonic stages, it is possible that Fd4 and Fd5 have partially redundant roles in specifying U4-U5 motor neurons, similar to the mammalian FOXP protein in GABAergic spiny neuron

specification (*Ahmed et al., 2024*). We hypothesize that the highly conserved Forkhead DNA-binding domain of Fd4 and Fd5 is required to activate Eve expression, but less well-conserved domains may regulate Fd4-specific and Fd5-specific function.

## How many 'neuroblast identity' genes exist?

Neuroblast lineages can be labeled by specific gal4 or split-gal4 drivers (*Lacin and Truman, 2016*; *Soffers et al., 2025*), documenting the potential lineage-specific gene expression. However, few genes have been identified that are expressed in single neuroblast lineages. In the *Drosophila* brain, the HDTF Orthodenticle (Otd; FlyBase: Oc) is expressed in a single neuroblast (LalV1) that generates central complex neurons, and loss of Otd transforms the neuroblast into a different neuroblast (ALad1) that generates olfactory projection neurons (*Sen et al., 2014*). Thus, Otd can be considered a neuroblast identity gene. Similarly, the NB5-6 lineage is the only neuroblast labeled by the HDTF Lbe, and misexpression of Lbe ubiquitously in other lineages also leads to the ectopic production of NB5-6-specific peptidergic lineage marker neurons (*Baumgardt et al., 2009*; *Gabilondo et al., 2016*). Thus, Otd and Lbe may join Fd4 as neuroblast identity genes that perform the same function: translating transient spatial cues that specify single neuroblasts into the permanent expression of lineage-specific terminal selector genes. Our findings raise the possibility that every neuroblast lineage may express its own neuroblast identity gene; alternatively, early-forming neuroblasts like NB7-1 and NB5-6 have the longest lineages and may require lineage identity genes to maintain neuroblast identity over the length of these lineages. Advances in single-cell RNA sequencing may reveal additional lineage-specific neuroblast identity genes.

## Methods

### Fly genetics

*eg-gal4* (RRID:BDSC_8758); *en-gal4* (*Schmid et al., 1999*); *lbe(K)-gal4* (*Baumgardt et al., 2009*); *sca-gal4* (Doe lab); *UAS-IVS-myr::GFP* (RRID:BDSC_32198); *UAS-myr::sfGFP* (RRID:BDSC_62127); *UAS-IVS-myr::smGdP-HA* (RRID:BDSC_62145); *UAS-RedStinger* (RRID:BDSC_8547); *vnd-GFP-FPTB* (RRID:BDSC_93583). *fd4* and *fd5* alleles: *Df(3R)BSC493/TM6C* (*fd4* and *fd5* deficiency) (RRID:BDSC_24997); *fd4^{5nt}/TM6B*, *fd5^{1nt}/TM6B*, and *UAS-fd4* were gifts from C Estella (Universidad Autónoma de Madrid, Madrid, Spain). All newly generated fly lines will be sent to the Bloomington Drosophila Stock Center (https://bdsc.indiana.edu/) for distribution to the public.

### Generation of *fd4/fd5* mutant

We used CRISPR to generate *fd4-gal4* by replacing *fd4* ORF with gal4 with the pHD-DsRed (Addgene plasmid #51434; http://n2t.net/addgene:51434; RRID:Addgene_51434) (*Gratz et al., 2014*), which also contained 1 kb of homologous arms up- and downstream of ORF for homology-directed repair (HDR). Two gRNAs (AACATTGTGTAATAATGCCC and TAGGATTCTCGCGAGGGCCG) were used to remove *fd4* ORF and cloned into pCFD4-U6:1_U6:3tandemgRNAs (Addgene plasmid #49411; http://n2t.net/addgene:49411; RRID:Addgene_49411) (*Port et al., 2014*). The gRNA and HDR constructs were co-injected into the recombined *Actin5C-Cas9.P; fd5^{1nt}/TM6B* flies by Rainbow Transgenic Flies, Inc (Camarillo, CA, USA).

### Antibody staining and imaging

Embryos were fixed and stained as previously described (*Grosskortenhaus et al., 2005*). Primary antibodies used were: rabbit anti-Cas (*Mellerick et al., 1992*), 1:1000 (Doe lab); rabbit anti-Corazonin, 1:2000 (*Isshiki et al., 2001*); guinea pig anti-Dbx, 1:200 (Doe lab); rat anti-Elav, 1:100 (DSHB, RRID:AB_528218); mouse anti-En, 5 µg/mL (DSHB, RRID:AB_528224); guinea pig anti-Eve, 1:200 (Desplan Lab, NYU, New York, NY, USA); mouse anti-Eve[2B8], 5 µg/mL (DSHB, RRID:AB_528230); rabbit anti-Eve, 1:250 (Doe lab); guinea pig anti-Fd4, 5 µg/mL (Doe lab); DyLight 488-conjugated goat anti-GFP, 1:400 (Novus Biologicals, Centennial, CO, USA); chicken anti-GFP, 1:1000 (Aves Labs, RRID:AB_2734732); mouse anti-Hb [F18-1G10.2], 1:200 (Abcam, Waltham, MA, USA); rabbit anti-Hb, 1:200 (*Tran and Doe, 2008*); rat anti-Ind, 1:100 (*Weiss et al., 1998*); rat anti-Nkx6, 1:500 (*Broihier et al., 2004*); rabbit anti-pMad [EP823Y], 1:300 (Abcam, Waltham, MA, USA); mouse anti-Prospero monoclonal purified IgG, 1:1000 (Doe lab); guinea pig anti-Runt, 1:1000 (*Sullivan et al., 2019*);

rat anti-Serotonin [YC5/45], 1:100 (Accurate Chemical & Scientific Corporation, Carle Place, NY, USA); rat anti-Tm1[MAC141], 1:500 (Abcam, Waltham, MA, USA); mouse anti-Wg, 5 μg/mL (DSHB, RRID:AB_528512); and rabbit anti-Worniu, 1:1000 (Doe lab). Secondary antibodies used were: DyLight 405, Alexa Fluor 488, Alexa Fluor 555, Rhodamine Red-X(RRX), or Alexa Fluor 647-conjugated Affini-Pure donkey anti-IgG (Jackson ImmunoResearch, West Grove, PA, USA). The samples were mounted in 90% glycerol with Vectashield (Vector Laboratories, Burlingame, CA, USA). Images were captured with a Zeiss LSM 800 confocal microscope with a z-resolution of 0.5 μm and processed using Imaris (Oxford Instruments plc, UK). Figures were assembled in Adobe Illustrator (Adobe, San Jose, CA, USA).

## Acknowledgements

We thank C Estella for flies, C Desplan, and J Skeath for antibodies, and Nathan Anderson, Austin Seroka, and Stefan Thor for comments on the manuscript. pHD-DsRed was a gift from Kate O'Connor-Giles, and pCFD4-U6:1_U6:3tandemgRNAs was a gift from Simon Bullock; both were obtained from Addgene. The rat anti-Elav, mouse anti-Eve[2B8], and mouse anti-Wg antibodies were obtained from the Developmental Studies Hybridoma Bank, created by the NICHD of the NIH and maintained at The University of Iowa, Department of Biology, Iowa City, IA 52242. Fly stocks obtained from the Bloomington Drosophila Stock Center (NIH P40OD018537) were used in this study. Funding was provided by HHMI (CQD and S-LL). The article submitted together with this notice is subject to the Immediate Access to Research policy of the Howard Hughes Medical Institute ('HHMI'). In accordance with this policy: (i) a preprint of this article either has been, or will be, deposited on a preprint server under a Creative Commons Attribution 4.0 International (CC BY 4.0) license and (ii) an additional author-published revised version of this article incorporating peer review feedback and/or new results or analysis either has been, or prior to journal publication will be, deposited on a preprint server under a CC BY 4.0 license. In addition, a nonexclusive CC BY 4.0 license to this article has been granted to the public and HHMI has a sublicensable, nonexclusive license to this article.

## Additional information

### Funding

| Funder | Grant reference number | Author |
| --- | --- | --- |
| Howard Hughes Medical Institute | | Chris Q Doe |
| National Institute of Health Sciences | HD27056 | Chris Q Doe |

The funders had no role in study design, data collection and interpretation, or the decision to submit the work for publication.

### Author contributions

Sen-Lin Lai, Conceptualization, Resources, Data curation, Formal analysis, Validation, Investigation, Visualization, Methodology, Writing – original draft, Writing – review and editing; Chris Q Doe, Conceptualization, Supervision, Funding acquisition, Writing – original draft, Project administration, Writing – review and editing

### Author ORCIDs

Sen-Lin Lai ⓘ https://orcid.org/0000-0002-7531-283X
Chris Q Doe ⓘ https://orcid.org/0000-0001-5980-8029

Reviewer #1 (Public review): https://doi.org/10.7554/eLife.109188.3.sa1
Reviewer #3 (Public review): https://doi.org/10.7554/eLife.109188.3.sa2
Author response: https://doi.org/10.7554/eLife.109188.3.sa3

## Additional files

### Supplementary files
MDAR checklist

### Data availability
We made a few several new fly lines, which are listed in Methods. These fly lines will be made publicly available from our lab on request or from the Bloomington Drosophila stock center (https://bdsc.indiana.edu/) for distribution. All other reagents were previously published.

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
