## [Editor Report · eLife Assessment]

This **important** study focuses on the molecular mechanisms underlying the generation of neuronal diversity. Taking advantage of a well-defined neuroblast lineage in Drosophila, the authors provide **convincing** evidence that two transcription factors of the conserved forkhead box (FOX) family offer a mechanistic link between transient spatial cues that specify neuroblast identity and terminal selector genes that define post-mitotic neuron identity. The findings will be of interest to developmental neurobiologists.

---

## [Referee Report · Reviewer #1 (Public review)]

Summary:

Lai and Doe address the integration of spatial information with temporal patterning and genes that specify cell fate. They identify the Forkhead transcription factor Fd4 as a lineage-restricted cell fate regulator that bridges transient spatial transcription factors to terminal selector genes in the developing Drosophila ventral nerve cord. The experimental evidence convincingly demonstrates that Fd4 is both necessary for late-born NB7-1 neurons, but also sufficient to transform other neural stem cell lineages toward the NB7-1 identity. This work addresses an important question that will be of interest to developmental neurobiologists: How cell identities defined by initial transient developmental cues can be maintained in the progeny cells, even if the molecular mechanism remains to be investigated. In addition, the study proposes a broader concept of lineage identity genes that could be utilized in other lineages and regions in the Drosophila nervous system and in other species.

Strengths:

While the spatial factors patterning the neuroepithelium to define the neuroblast lineages in the Drosophila ventral nerve cord are known, these factors are sometimes absent or not required during neurogenesis. In the current work, Lai and Doe identified Fd4 in the NB7-1 lineage that bridges this gap and explains how NB7-1 neurons are specified after Engrailed (En) and Vnd cease their expression. They show that Fd4 is transiently co-expressed with En and Vnd and are present in all nascent NB7-1 progenies. They further demonstrate that Fd4 is required for later-born NB7-1 progenies and sufficient for the induction of NB7-1 markers (Eve and Dbx) while repressing markers of other lineages when force-expressed in neural progenitors, e.g. in the NB5-6 lineage and in the NB7-3 lineage. They also demonstrate that, when Fd4 is ectopically expressed in NB7-3 and NB5-6 lineages, this leads to the ectopic generation of dorsal muscle-innervating neurons. The inclusion of functional validation using axon projections demonstrates that the transformed neurons acquire appropriate NB7-1 characteristics beyond just molecular markers. Quantitative analyses are thorough and well-presented for most experiments.

Original weaknesses and potential extensions:

(1) While Fd4 is required and sufficient for several later-born NB7-1 progeny features, a comparison between early-born (Hb/Eve) and later-born (Run/Eve) appears missing for pan-progenitor gain of Fd4 (with sca-Gal4; Figure 4) and for the NB7-3 lineage (Figure 6). Having a quantification for both could make it clearer whether Fd4 preferentially induces later-born neurons or is sufficient for NB7-1 features without temporal restriction.

(2) Fd4 and Fd5 are shown to be partially redundant, as Fd4 loss of function alone does not alter the number of Eve+ and Dbx+ neurons. This information is critical and should be included in Figure 3.

(3) Several observations suggest that lineage identity maintenance involves both Fd4-dependent and Fd4-independent mechanisms. In particular, the fact that fd4-Gal4 reporter remains active in fd4/fd5 mutants even after Vnd and En disappear indicates that Fd4's own expression, a key feature of NB7-1 identity, is maintained independently of Fd4 protein. This raises questions about what proportion of lineage identity features require Fd4 versus other maintenance mechanisms, which deserves discussion.

(4) Similarly, while gain of Fd4 induces NB7-1 lineage markers and dorsal muscle innervation in NB5-6 and NB7-3 lineages, drivers for the two lineages remain active despite the loss of molecular markers, indicating some regulatory elements retain activity consistent with their original lineage identity. It is therefore important to understand the degree of functional conversion in the gain-of-function experiments. Sparse labeling of Fd4 overexpressing NB5-6 and NB7-3 progenies, as what was done in Seroka and Doe (2019) would be an option.

(5) The less-penetrant induction of Dbx+ neurons in NB5-6 with Fd4-overexpression is interesting. It might be worth discussing whether it is a Fd4 feature or a NB5-6 feature by examining Dbx+ neuron number in NB7-3 with Fd4-overexpression.

(6) It is logical to hypothesize that spatial factors specify early-born neurons directly so only late-born neurons require Fd4, but it was not tested. The model would be strengthened by examining whether Fd4-Gal4-driven Vnd rescues the generation of later-born neurons in fd4/fd5 mutants.

(7) It is mentioned that Fd5 is not sufficient for the NB7-1 lineage identity. The observation is intriguing in how similar regulators serve distinct roles, but the data are not shown. The analysis in Figure 4 should be performed for Fd5 as supplemental information.

Comments on latest version:

We appreciate the thorough revision and the detailed point-by-point responses. Overall, the updated manuscript has addressed the main issues we raised previously, especially around the potential potency differences of Fd4 along the birth order axis and possible redundancy with Vnd in early-born neurons. The additional data are convincing and presented clearly, with figures and supplements that are informative and appropriately labeled.

We noticed one remaining point that could be considered, the necessary-and-sufficient phrasing for Fd4 regulating NB7-1 fates. Given the possible redundancy among Fd4/5 and Vnd and the fact that early-born outputs (U1-3, Figure 3F) are not dependent on Fd4/5, we suggest revising this claim and either (a) limit the claim to necessary and sufficient for late-born NB7-1 progeny identity, or (b) frame Fd4 as sufficient for NB7-1 program induction while being required but redundant (e.g., with Vnd) for early-born features, rather than universally necessary/sufficient across the entire lineage output.

Regarding the lack of phenotype of single Fd4/5 mutants and Fd5 gain of function, we still encourage the authors to include the fd4 and fd5 single-mutant negative results as a brief supplemental item (e.g., a representative panel plus a simple quantification on Eve and Dbx would be sufficient). This would strengthen transparency, remove "data not shown" statements that are not necessary when these data can be presented as supplementary data with no space limitation, and make it easier for readers to evaluate redundancy claims.

Overall, we view the work as substantially complete and appreciate its contribution and conceptual framing. We have updated our public review to reflect the current version and the authors' efforts to address the major points raised in the prior round.

---

## [Referee Report · Reviewer #3 (Public review)]

The goal of the work is to establish the linkage between the spatial transcription factors (STF's) that function transiently to establish the identities of the individual NBs and the terminal selector genes (typically homeodomain genes) that appear in the new-born post-mitotic neurons. How is the identity of the NB maintained and carried forward after the spatial genes have faded away? Focusing on a single neuroblast (NB 7-1), the authors present evidence that the fork-head transcription factor, fd4, provides a bridge linking the transient spatial cues that initially specified neuroblast identity with the terminal selector genes that establish and maintain the identity of the stem cell's progeny.

The study is systematic, concise and takes full advantage of 40+ years of work on the molecular players that establish neuronal identities in the Drosophila CNS. In the embryonic VNC, fd4 is expressed only in the NB 7-1 and its lineage. They show that Fd4 appears in the NB while the latter is still expressing the Spatial Transcription Factors and continues after the expression of the latter fades out. Fd4 is maintained through the early life of the neuronal progeny but then declines as the neurons turn on their terminal selector genes. Hence, fd4 expression is compatible with it being a bridging factor between the two sets of genes.

Experimental support for the "bridging" role of Fd4 comes from set of loss-of-function and gain-of-function manipulations. The loss of function of fd4, and the partially redundant gene fd5, from lineage 7-1 does not affect the size of the lineage, but terminal markers of late-born neuronal phenotypes, like Eve and Dbx, are reduced or missing. By contrast, ectopic expression of fd4, but not fd5, results in ectopic expression of the terminal markers eve and dbx throughout diverse VNC lineages.

A detailed test of fd4's expression was then carried out using lineages 7-3 and 5-6, two well characterized lineages in Drosophila. Lineage 7-3 is much smaller that 7-1 and continues to be so when subjected to fd4 misexpression. However, under the influence of ectopic fd4 expression, the lineage 7-3 neurons lost their expected serotonin and corazonin expression and showed Eve expression as well as motoneuron phenotypes that partially mimic the U motoneurons of lineage 7-1.

Ectopic expression of Fd4 also produced changes in the 5-6 lineage. Expression of apterous, a feature of lineage 5-6 was suppressed, and expression of the 7-1 marker, Eve, was evident. Dbx expression was also evident in the transformed 5-6 lineages but extremely restricted as compared to a normal 7-1 lineage. Considering the partial redundancy of fd4 and fd5, it would have been interesting to express both genes in the 5-6 lineage. The anatomical changes that are exhibited by motoneurons in response to fd4 expression confirms that these cells do, indeed, show a shift in their cellular identity.

Comments on revisions:

The authors adequately addressed all of the issues that I had with the original submission.

Their responses to the other reviewers are also well-reasoned

---

## [Author Response]

The following is the authors’ response to the original reviews.

**Public Reviews:**

**Reviewer #1 (Public review):**
Lai and Doe address the integration of spatial information with temporal patterning and genes that specify cell fate. They identify the Forkhead transcription factor Fd4 as a lineage-restricted cell fate regulator that bridges transient spatial transcription factors to terminal selector genes in the developing Drosophila ventral nerve cord. The experimental evidence convincingly demonstrates that Fd4 is both necessary for lateborn NB7-1 neurons, but also sufficient to transform other neural stem cell lineages toward the NB7-1 identity. This work addresses an important question that will be of interest to developmental neurobiologists: How can cell identities defined by initial transient developmental cues be maintained in the progeny cells, even if the molecular mechanism remains to be investigated? In addition, the study proposes a broader concept of lineage identity genes that could be utilized in other lineages and regions in the Drosophila nervous system and in other species.

Thanks for the accurate summary and positive comments!

While the spatial factors patterning the neuroepithelium to define the neuroblast lineages in the Drosophila ventral nerve cord are known, these factors are sometimes absent or not required during neurogenesis. In the current work, Lai and Doe identified Fd4 in the NB7-1 lineage that bridges this gap and explains how NB7-1 neurons are specified after Engrailed (En) and Vnd cease their expression. They show that Fd4 is transiently co-expressed with En and Vnd and is present in all nascent NB7-1 progenies. They further demonstrate that Fd4 is required for later-born NB7-1 progenies and sufficient for the induction of NB7-1 markers (Eve and Dbx) while repressing markers of other lineages when force-expressed in neural progenitors, e.g., in the NB56 lineage and in the NB7-3 lineage. They also demonstrate that, when Fd4 is ectopically expressed in NB7-3 and NB5-6 lineages, this leads to the ectopic generation of dorsal muscle-innervating neurons. The inclusion of functional validation using axon projections demonstrates that the transformed neurons acquire appropriate NB7-1 characteristics beyond just molecular markers. Quantitative analyses are thorough and well-presented for all experiments.

Thanks for the positive comments!

(1) While Fd4 is required and sufficient for several later-born NB7-1 progeny features, a comparison between early-born (Hb/Eve) and later-born (Run/Eve) appears missing for pan-progenitor gain of Fd4 (with sca-Gal4; Figure 4) and for the NB7-3 lineage (Figure 6). Having a quantification for both could make it clearer whether Fd4 preferentially induces later-born neurons or is sufficient for NB7-1 features without temporal restriction.

We quantified the percentage of Hb+ and Runt+ cells among Eve+ cells with sca-gal4, and the results are shown in Figure 4-figure supplement 1. We found that the proportion of early-born cells is slightly reduced but the proportion of later-born cells remain similar. Interestingly, we also found a subset of Eve+ cells with a mixed fate (Hb+Runt+) but the reason remains unclear.

(2) Fd4 and Fd5 are shown to be partially redundant, as Fd4 loss of function alone does not alter the number of Eve+ and Dbx+ neurons. This information is critical and should be included in Figure 3.

Because every hemisegment in an fd4 single mutant is normal, we just added it as the following text: “In fd4 mutants, we observe no change in the number of Eve+ neurons or Dbx+ neurons (n=40 hemisegments).”

(3) Several observations suggest that lineage identity maintenance involves both Fd4dependent and Fd4-independent mechanisms. In particular, the fact that fd4-Gal4 reporter remains active in fd4/fd5 mutants even after Vnd and En disappear indicates that Fd4's own expression, a key feature of NB7-1 identity, is maintained independently of Fd4 protein. This raises questions about what proportion of lineage identity features require Fd4 versus other maintenance mechanisms, which deserves discussion.

We agree, thanks for raising this point. We add the following text to the Discussion. “Interestingly, the fd4 fd5 mutant maintains expression of fd4:gal4, suggesting that the fd4/fd5 locus may have established a chromatin state that allows “permanent” expression in the absence of Vnd, En, and Fd4/Fd5 proteins.”

(4) Similarly, while gain of Fd4 induces NB7-1 lineage markers and dorsal muscle innervation in NB5-6 and NB7-3 lineages, drivers for the two lineages remain active despite the loss of molecular markers, indicating some regulatory elements retain activity consistent with their original lineage identity. It is therefore important to understand the degree of functional conversion in the gain-of-function experiments. Sparse labeling of Fd4 overexpressing NB5-6 and NB7-3 progenies, as was done in Seroka and Doe (2019), would be an option.

We agree it is interesting that the NB7-3 and NB5-6 drivers remain on following Fd4 misexpression. To explore this, we used sca-gal4 to overexpress Fd4 and observed that Lbe expression persisted while Eg was largely repressed (Author response image 1). The results show that Lbe and Eg respond differently to Fd4. A non-mutually exclusive possibility is that the continued expression of lbe-Gal4 UAS-GFP or eg-Gal4 UAS-GFP may be due to the lengthy perdurance of both Gal4 and GFP.

(5) The less-penetrant induction of Dbx+ neurons in NB5-6 with Fd4-overexpression is interesting. It might be worth the authors discussing whether it is an Fd4 feature or an NB56 feature by examining Dbx+ neuron number in NB7-3 with Fd4-overexpression.

In the NB7-3 lineages misexpressing Fd4, only 5 lineages generated Dbx+ cells (0.1±0.4, n=64 hemisegments), suggesting that the low penetrance of Dbx+ induction is an intrinsic feature of Fd4 rather than lineage context. We have added this information in the results section.

(6) It is logical to hypothesize that spatial factors specify early-born neurons directly, so only late-born neurons require Fd4, but it was not tested. The model would be strengthened by examining whether Fd4-Gal4-driven Vnd rescues the generation of laterborn neurons in fd4/fd5 mutants.

When we used en-gal4 driver to express UAS-vnd in the fd4/fd5 mutant background, we found an average 7.4±2.2 Eve+ cells per hemisegment (n=36), significantly higher than fd4/fd5 mutant alone (3.9±0.8 cells, n=52, p=2.6x10^-11^) (Figure 3J). In addition, 0.2±0.5 Eve+ cells were ectopic Hb+ (excluding U1/U2), indicating that Vnd-En integration is sufficient to generate both early-born and late-born Eve+ cells in the fd4/fd5 mutants. We have added the results to the text.

(7) It is mentioned that Fd5 is not sufficient for the NB7-1 lineage identity. The observation is intriguing in how similar regulators serve distinct roles, but the data are not shown. The analysis in Figure 4 should be performed for Fd5 as supplemental information.

Thanks for the suggestion. Because the results are exactly the same as the wild type, we don’t think it is necessary to provide an additional images or analysis as supplemental information.

**Reviewer #2 (Public review):**
Via a detailed expression analysis, they find that Fd4 is selectively expressed in embryonic NB7-1 and newly born neurons within this lineage. They also undertake a comprehensive genetic analysis to provide evidence that fd4 is necessary and sufficient for the identity of NB7-1 progeny.

Thanks for the accurate summary!

The analysis is both careful and rigorous, and the findings are of interest to developmental neurobiologists interested in molecular mechanisms underlying the generation of neuronal diversity. Great care was taken to make the figures clear and accessible. This work takes great advantage of years of painstaking descriptive work that has mapped embryonic neuroblast lineages in Drosophila.

Thanks for the positive comments!

The argument that Fd4 is necessary for NB7-1 lineage identity is based on a Fd4/Fd5 double mutant. Loss of fd4 alone did not alter the number of NB7-1-derived Eve+ or Dbx+ neurons. The authors clearly demonstrate redundancy between fd4 and fd5, and the fact that the LOF analysis is based on a double mutant should be better woven through the text.The authors generated an Fd5 mutant. I assume that Fd5 single mutants do not display NB7-1 lineage defects, but this is not stated. The focus on Fd4 over Fd5 is based on its highly specific expression profile and the dramatic misexpression phenotypes. But the LOF analysis demonstrates redundancy, and the conclusions in the abstract and through the results should reflect the existence of Fd5 in the conclusions of this manuscript.

We agree, and have added new text to clarify the single mutant phenotypes (there are none) and the double mutant phenotype loss of NB7-1 molecular and morphological features. The following text is added to the manuscript: “Not surprisingly, we found that fd4 single mutants or fd5 single mutants had no phenotype (Eve+ neurons were all normal). Thus, to assess their roles, we generated a fd4 and fd5 double mutant. Because many Eve+ and Dbx+ cells are generated outside of NB7-1 lineage, it was also essential to identify the Eve+ or Dbx+ cells within NB7-1 lineage in wild type and fd4 mutant embryos. To achieve this, we replaced the open reading frame of fd4 with gal4 (called fd4-gal4) (see Methods); this stock simultaneously knocked out both fd4 and fd5 (called fd4/fd5 mutant hereafter) while specifically labeling the NB7-1 lineage. For the remainder of this paper we use the fd4/fd5 double mutant to assay for loss of function phenotypes.”

It is notable that Fd4 overexpression can rewire motor circuits. This analysis adds another dimension to the changes in transcription factor expression and, importantly, demonstrates functional consequences. Could the authors test whether U4 and U5 motor axon targeting changes in the fd4/fd5 double mutant? To strengthen claims regarding the importance of fd4/fd5 for lineage identity, it would help to address terminal features of U motorneuron identity in the LOF condition.

Thanks for raising this important point. We examined the axon targeting on body wall muscles in both wild type and in fd4/fd5 mutant background and added the results in Figure 3-figure supplement 2. We found that the axon targeting in the late-born neuron region (LL1) is significantly reduced, suggesting that the loss of late-born neurons in fd4/fd5 mutant leads to the absence of innervation of corresponding muscle targets.

**Reviewer #3 (Public review):**
The goal of the work is to establish the linkage between the spatial transcription factors (STFs) that function transiently to establish the identities of the individual NBs and the terminal selector genes (typically homeodomain genes) that appear in the newborn postmitotic neurons. How is the identity of the NB maintained and carried forward after the spatial genes have faded away? Focusing on a single neuroblast (NB 7-1), the authors present evidence that the fork-head transcription factor, fd4, provides a bridge linking the transient spatial cues that initially specified neuroblast identity with the terminal selector genes that establish and maintain the identity of the stem cell's progeny.

Thanks for the positive comments!

The study is systematic, concise, and takes full advantage of 40+ years of work on the molecular players that establish neuronal identities in the Drosophila CNS. In the embryonic VNC, fd4 is expressed only in the NB 7-1 and its lineage. They show that Fd4 appears in the NB while the latter is still expressing the Spatial Transcription Factors and continues after the expression of the latter fades out. Fd4 is maintained through the early life of the neuronal progeny but then declines as the neurons turn on their terminal selector genes. Hence, fd4 expression is compatible with it being a bridging factor between the two sets of genes.

Thanks for the accurate summary!

Experimental support for the "bridging" role of Fd4 comes from a set of loss-of-function and gain-of-function manipulations. The loss of function of Fd4, and the partially redundant gene Fd5, from lineage 7-1 does not aoect the size of the lineage, but terminal markers of late-born neuronal phenotypes, like Eve and Dbx, are reduced or missing. By contrast, ectopic expression of fd4, but not fd5, results in ectopic expression of the terminal markers eve and Dbx throughout diverse VNC lineages.

Thanks for the accurate summary!

A detailed test of fd4's expression was then carried out using lineages 7-3 and 5-6, two well-characterized lineages in Drosophila. Lineage 7-3 is much smaller than 7-1 and continues to be so when subjected to fd4 misexpression. However, under the influence of ectopic Fd4 expression, the lineage 7-3 neurons lost their expected serotonin and corazonin expression and showed Eve expression as well as motoneuron phenotypes that partially mimic the U motoneurons of lineage 7-1.

Thanks for the positive comments!

Ectopic expression of Fd4 also produced changes in the 5-6 lineage. Expression of apterous, a feature of lineage 5-6, was suppressed, and expression of the 7-1 marker, Eve, was evident. Dbx expression was also evident in the transformed 5-6 lineages, but extremely restricted as compared to a normal 7-1 lineage. Considering the partial redundancy of fd4 and fd5, it would have been interesting to express both genes in the 5-6 lineage. The anatomical changes that are exhibited by motoneurons in response to Fd4 expression confirm that these cells do, indeed, show a shift in their cellular identity.

We appreciate the positive comments. We agree double misexpression of Fd4 and Fd5 might give a stronger phenotype (as the reviewer says) but the lack of this experiment does not change the conclusions that Fd4 can promote NB7-1 molecular and morphological aspects at the expense of NB5-6 molecular markers.

**Recommendations for the authors:**

**Reviewer #2 (Recommendations for the authors):**
The title of Figure 4 may be intended to include the term "Widespread", not "Wild spread". (Though the expansion of the Eve and Dbx with Fd4 is quite remarkable…).

Done!

**Reviewer #3 (Recommendations for the authors):**
(1) Line 138. Is part of the sentence missing? Did the authors mean to say "that fd5 is coexpressed with fd4 in NB7-1 and its .....".

Done!

(2) ln 237: In trying to explain the "U-like" phenotype of the transformed motoneurons in lineage 7-3, the authors speculate that "perhaps their late birth did not give them time to extend to the most distant dorsal muscles ". It is very difficult to convince a motoneuron to stop growing in the absence of a target! An alternate possibility is that since there is only one or two U neurons made instead of the normal five, the growing motoneuron has enough information to direct them to the dorsal domain, but they lack the specification that allows them to recognize a specific muscle target.

We agree there are additional possibilities, and now update the text to say: “We observed that these transformed neurons did not innervate the dorsal muscles, perhaps their late birth did not give them time to extend to the most distant dorsal muscles, or they were incompletely specified.”

(3) In the References, I think that the Anderson et al. reference should also include "BioRxiv" before the DOI.

Done!

(4) Figure 6A for wild-type 7-3 lineage. The corazonin expression appears to be expressed in EW2 as well as EW3. This should be explained.

We agree it looks that way, due to the 3D rotation used; we now replace it with a more representative image. Note that our quantification always shows a single Cor+ neuron per hemisegment.

(5) Figure 7: Issues of terminology. The designation of "longitudinal" for muscles is traditionally in reference to the body axis, such as the Dorsal Longitudinal Muscles (DLM) of the adult thorax. The "longitudinal" muscles in the figure are really "transverse" muscles. I also suggest using "axon" or "neurites" rather than "filament". For the middle and bottom parts of E and F, are these lateral and ventral views? They should be designated as such.

Thanks, we agree and have made the changes, using Axon instead of Filament, and labeling the views (lateral and ventro-lateral).